# Mesenteric Ischemia in a Splenectomized Patient with Auto-Immune Hemolytic Anemia: Case Report

**DOI:** 10.3390/medicina59071325

**Published:** 2023-07-18

**Authors:** Sinthia Vidal-Cañas, Cristian Zuñiga-Jaramillo, Esteban Artunduaga-Cañas, Valentina Pérez-Garay, Yamil Liscano

**Affiliations:** Grupo de Investigación en Salud Integral (GISI), Departamento Facultad de Salud, Universidad Santiago de Cali, Cali 5183000, Colombia; xonara16@gmail.com (C.Z.-J.); esteban.artunduaga01@uceva.edu.co (E.A.-C.); valeperez239@gmail.com (V.P.-G.)

**Keywords:** mesenteric ischemia, mesenteric vein, hemolytic anemia, venous thrombosis

## Abstract

Mesenteric ischemia is a serious complication that can occur after splenectomy for hemolytic anemia, potentially leading to lifelong intestinal problems such as ischemia and/or portal hypertension. We present the case of a 33-year-old man with a history of autoimmune hemolytic anemia and splenectomy who developed mesenteric ischemia. The patient experienced abdominal pain and diarrhea, and imaging studies revealed mesenteric vein thrombosis. Surgical intervention confirmed the diagnosis. This case significantly contributes to the existing literature by providing insights into the occurrence of mesenteric ischemia in younger individuals with predisposing factors, as well as its clinical presentation, diagnostic challenges, and severity. Moreover, it has implications for the future diagnosis and management of long-term mesenteric ischemia in patients who have undergone splenectomy for hemolytic anemia.

## 1. Introduction

Mesenteric vein thrombosis is a serious and potentially fatal complication that can occur as a result of splenectomy in patients with established hemolytic anemia. It can lead to acute intestinal infarction and extrahepatic portal hypertension and is associated with various systemic and local risk factors for portal-splenic mesenteric venous thrombosis (PSMVT) [1]. The prevalence of PSMVT ranges from 5% to 50% [1]. Iolascon et al. (2017) [2] have reported an increased risk of both early and late venous and arterial insufficiency from the portal and splenic veins following splenectomy. A Danish registry study of patients who underwent splenectomy between 1996 and 2005 showed that the long-term (>1 year) risk of venous thromboembolism remained approximately three times higher in these patients compared to the general population [2]. The mechanisms underlying portal vein thrombosis (PVT) after splenectomy are still unknown, but they are believed to be related to the classic Virchow triad, involving venous stasis, hypercoagulability, and endothelial damage. Spleen resection reduces portal blood flow, leading to turbulence and stasis in the stump, resulting in increased diameters of the portal vein and splenic vein, and eventually leading to thrombosis [3]. The onset of mesenteric vein thrombosis is often less sudden compared to other forms of mesenteric ischemia, and the pain is typically dull. Over 75% of patients report experiencing at least two days of pain before seeking medical attention [4]. It is important to highlight that all the information gathered and utilized in this case report was de-identified. Any patient-specific information that could potentially identify the patient was removed. The purpose of presenting this case is to contribute to the understanding and clinical management of mesenteric ischemia while preserving the patient’s privacy and confidentiality. Here, we present a case of mesenteric ischemia caused by mesenteric venous thrombosis secondary to splenectomy. There is limited information available in the literature regarding patients with splenectomy who develop mesenteric ischemia due to mesenteric venous thrombosis.

## 2. The Case Description

On 4 February 2023, a 33-year-old Colombian man was referred to our institution. He had been experiencing colicky abdominal pain in the right hypochondrium, particularly after consuming cholecystokinin-rich food, for the past month. His medical history includes a splenectomy seven years ago due to autoimmune hemolytic anemia and a diagnosis of insulin-requiring type 2 diabetes mellitus three years ago. Two days prior to admission, he developed intense non-radiating colicky abdominal pain in the upper abdomen, accompanied by nausea and multiple episodes of bilious vomiting. He also experienced abdominal distention (Figure 1) and diarrhea, which later progressed to the absence of stools. He reported an unmeasured increase in temperature but denied any other associated symptoms. This episode marked the first occurrence of severe abdominal pain in his life. His medication regimen includes steroids, insulin glargine, and insulin glulisine, although he has not been compliant with treatment. He denied having any allergies and mentioned occasional use of psychoactive substances such as marijuana and cocaine. He has been a heavy smoker for 16 years, consuming two packs of cigarettes per day, which puts him at a high risk of developing chronic obstructive pulmonary disease (COPD). The splenectomy he underwent seven years ago was successful without any complications. On physical examination, the patient appeared diaphoretic, and the following physical signs were noted: blood pressure of 115/82 mmHg, heart rate of 102 beats per minute, respiratory rate of 16 breaths per minute, and body temperature of 36.4 °C. His abdomen was distended, and a well-healed incision was observed in the left upper quadrant. Palpation of the right hypochondrium and peri-umbilical region elicited pain, but there were no palpable masses or organ enlargement, and no signs of peritoneal irritation were present.

Upon admission, several paraclinical tests were performed. A blood count revealed leukocytosis with an increase in neutrophils, mild normocytic anemia, and normal platelet levels. Total bilirubin was elevated at 1.29 mg/dL, primarily due to elevated direct bilirubin at 0.80 mg/dL. Blood urea nitrogen was measured at 16.3 mg/dL, and creatinine at 0.54 mg/dL. Electrolyte levels were within normal ranges, including chloride at 104 mmol/L, potassium at 3.8 mmol/L, and sodium at 140 mmol/L. On February 6, arterial blood gas analysis was conducted, which showed a lactic acid level of 20.2 mg/dL, pH of 7.4, pCO_2_ of 34 mmHg, PO_2_ of 73.4 mmol/L, bicarbonate (HCO_3−_) level of 22.4 nmol/L, base excess (BE) of −2.4, and oxygen saturation (O_2_SAT) of 95.1%. Glycosylated hemoglobin and ketone bodies were within normal limits. Blood and urine cultures yielded negative results. Coprology analysis revealed coffee-colored stool with liquid consistency, positive for occult blood, increased bacterial flora, and negative for intestinal parasites. Lipase level was measured at 0.55 (U/L), and the coagulation profile indicated slightly prolonged prothrombin time (PT), while activated partial thromboplastin time (aPTT) and international normalized ratio (INR) were within normal ranges (see Table 1).

Hepatobiliary ultrasound was performed, revealing an enlarged liver with a slight increase in echogenicity and no evidence of cystic or solid focal lesions. The intra- and extrahepatic biliary tracts appeared normal in caliber. The right hepatic lobe measured 175 mm. The gallbladder appeared distended, measuring 94 × 35 mm, with thickened walls measuring 4.5 mm. Low-level echoes were observed inside the gallbladder, occupying approximately 75% of its volume. Additionally, there was moderate ascites with clear and non-septate content. An abdominal series was also conducted, revealing the presence of pneumatosis intestinalis, interloop edema, and the footprint sign. Moreover, PA and lateral thoracic radiography showed a right pleural effusion.

The contrast abdominal (CT) revealed discrete alveolar opacities located subpleurally in the right lung base (Figure 2). The liver appeared hypodense with moderate fatty infiltration, without any observed masses or collections. Abundant ascitic fluid was present around the liver. The intra- and extrahepatic biliary tracts showed no dilation. The gallbladder was moderately distended, and hyperdense images were observed inside, indicating the presence of gallstones. The pancreas had no focal lesions, and gastric distension was noted. A discrete hiatal hernia was observed, along with marked mucosal thickening at the level of the greater curvature. The adrenal glands appeared normal, and the kidneys had smooth contours and homogeneous density, without any masses, obstructive signs, or stones. The bladder appeared normal. Intestinal loops showed distinction, with the presence of hydroaereal levels in the small intestine and areas of lumen narrowing in the distal loops. Air was present in the colic frame and rectum sigma. Discrete ascitic fluid was observed in the parietocolic drip and pelvic region.

Upon admission, the patient’s pharmacological management included intravenous administration of 10 mg/2 mL metoclopramide solution every 12 h, butylscopolamine + dipyrone 0.02 + 2.5 every 6 h, and ampicillin + sulbactam 3 g every 6 h. However, due to an unfavorable clinical progression and susceptibility to sepsis, the management was adjusted during the hospital stay. Cefepime 1 g was added every 6 h intravenously, along with a 2 mg loperamide tablet every 8 h orally for symptom relief, and for pain management, 50 mg-mL tramadol was taken every 6 h. Obtaining more specific tests such as angio-CT posed challenges due to limited economic resources and delays in the healthcare system’s response, which would have resulted in test results being available only after 1 week, while the patient required urgent imaging.

The patient’s symptoms worsened, presenting diffuse and disproportionate abdominal pain and signs of peritoneal irritation. Systemic inflammatory response symptoms developed, and the patient became hypotensive, requiring surgical intervention. Intraoperative findings revealed a 2500 cc intraperitoneal collection and mesenteric ischemia of the jejunum at 80 cm of the Treitz ligament, with approximately 2 m of ischemic loops. The mesentery and intestinal wall were thickened (Figure 3). The intraperitoneal fluid collection was drained and sent for cytology and pathology analysis. Ischemic bowel resection was performed, and damage control measures were implemented. The abdominal cavity was irrigated; a Bogota bag was placed; and the incision was closed. The patient tolerated the procedure, and the prognosis is dependent on the underlying pathology. In general, the prognosis for patients with severe mesenteric ischemic events is guarded, requiring continuous monitoring and treatment to prevent potential relapses and improve long-term outcomes. While other potential diagnoses such as intestinal obstruction with strangulation may be considered, the patient’s history of hemolytic anemia and splenectomy places him at higher risk for mesenteric ischemia.

## 3. Discussion

Mesenteric vein thrombosis, if left undiagnosed, has a fatality rate of 20% due to bowel ischemia. Patients with hemolytic anemia may have chronic thrombocytosis, making them more susceptible to thrombi and pulmonary emboli when red blood cell deformability decreases and blood viscosity increases after splenectomy. These factors can contribute to ischemia [5], as seen in this patient’s case. Splenectomy is likely associated with thrombus formation in large veins. Mesenteric vein thrombosis tends to affect the small intestines more frequently than the colon, with the ileum being the most commonly involved site (83%), followed by the jejunum (50–81%), colon (14%), and duodenum (8%) [6]. In this patient, surgical findings revealed ischemia in approximately 2 m of the jejunum. Drainage is affected, leading to an increase in venous pressure and leakage of fluid into the tissues. Consequently, deep edema of the intestinal wall is observed, which can result in submucosal layer hemorrhage. If the venous arcades and vasa recta are affected, causing complete venous occlusion of the entire intestinal wall, intestinal infarction can occur [7,8]. The patient’s abdominal series showed inter-loop edema, intestinal wall thickening, ascites, and right pleural effusion. Additionally, sequestration of fluid in the bowel lumen combined with massive bowel wall edema leads to relative hypovolemia and systemic hypotension. This reduction in arterial flow exacerbates ischemia [9], which explains the patient’s hypotension and signs of peritoneal irritation on the day of surgery.

Acute mesenteric venous thrombosis, like other forms of acute mesenteric ischemia, is characterized by colicky, periumbilical abdominal pain that persists for at least a few hours and is disproportionate to the physical examination findings initially. However, the pain is less abrupt and duller compared to other forms of mesenteric ischemia. Over 75% of patients report experiencing pain for at least two days before seeking medical attention. Abdominal distension and occult blood in the stool may be present, while signs of peritoneal inflammation such as rebound tenderness and guarding are typically absent. However, as bowel distention progresses, ischemia can occur, leading to absent bowel sounds and the development of peritoneal signs [10]. In this case, it took 2 days to perform the procedure, and the patient presented with a distended abdomen on physical examination, initially experiencing pain in the periumbilical region and absence of stool, which later progressed to diffuse pain and signs of peritoneal irritation. Plain abdominal radiographs are relatively nonspecific and may appear normal in 25% of patients. Findings suggestive of acute mesenteric ischemia include ileus with distended bowel loops and thickening of the bowel wall. The latter is particularly prominent in patients with acute mesenteric venous thrombosis and/or pneumatosis intestinalis [10].

Laparoscopic surgery has been associated with blood stasis and hypercoagulability of the portal circulation in previous studies [11,12]. Reduced portal and hepatic blood flow can cause endothelial damage and expose procoagulant tissue components [13,14]. In this patient’s case, the splenectomy performed via laparoscopy 7 years ago constituted a risk factor for thrombus formation. Additionally, the patient’s specific coagulation profile should be considered. Deficiency in protein S, Factor 5 Leiden mutation, and Prothrombin G2021A mutation can further increase the risk of thrombosis. These genetic alterations affect the normal functioning of coagulation factors and predispose individuals to abnormal clot formation [15,16].

In a case report by Wu et al. (2018), an 85-year-old man with a history of hypertension and atrial fibrillation presented with progressive periumbilical cramping pain of acute onset. The pain, which was not relieved by painkillers, occurred after food intake and was accompanied by cold sweating, nausea, and vomiting. A non-contrast abdominal CT scan was performed due to poor renal function, revealing no free air in the abdomen but a slight increase in air in the undistended small bowel loops. The patient also had gallbladder stones with slight strands of local fat, leading to a suspicion of cholecystitis. Antibiotics were administered, but due to worsening symptoms, surgery was performed, and mesenteric ischemia was discovered [17]. This case is similar to the patient described here, as they presented with a comparable clinical picture initially mistaken for cholelithiasis with cholecystitis.

Wu et al. (2018) [17] included 50 patients (40 men and 10 women) admitted to the surgical department between January 2007 and December 2008. Among these patients, eight cases of portal vein thrombosis (16%) were identified following splenectomy. The eight patients included four out of ten (40%) with myeloproliferative disease (MPD), three out of twelve (25%) with hemolytic anemia and spleen weighing more than 3000 g, and one out of ten (10%) who underwent splenectomy for hypersplenism. All patients had splenomegaly with a mean weight of 1540 g (range 460 to 3850 g). Symptoms at presentation included anorexia (87.5%) and abdominal pain (75%), and all cases had elevated D-dimer levels, leukocyte counts, and platelet counts. The diagnoses were confirmed using contrast-enhanced CT, and anticoagulation therapy was initiated immediately. One out of eight patients (12.5%) died due to progressive hepatocellular failure, while the others survived without clinical sequelae after a mean follow-up of 27 months [17]. In the patient’s case, continuous vomiting was indicative of anorexia, and the CT scan was partially helpful in making the diagnosis.

Regarding the clinical presentation, intense abdominal pain typically begins in the periumbilical region. However, when intestinal infarction occurs, signs of peritoneal irritation may develop. Notably, an abdominal series can provide supportive diagnostic information. After a few hours, patients may develop ascites due to fluid extravasation. Deep vein thrombosis is a common complication of splenectomy in patients with splenomegaly who already have a higher incidence of thrombotic disorders. Surgeons should be vigilant in making an early diagnosis through contrast-enhanced computed tomography and initiating immediate anticoagulation for optimal outcomes [18].

## 4. Conclusions

Mesenteric ischemia is a multifactorial event in which thrombosis due to splenectomy as a treatment for hemolytic anemia becomes a very frequent postoperative complication. This complication can manifest early or result in serious late complications for a lifetime, leading to intestinal problems such as ischemia and/or portal hypertension. Management of this condition requires prophylactic recommendations after the procedure, such as anticoagulants for a period of 3–6 months to decrease the incidence of thrombosis. However, there is a higher long-term risk of portal vein thrombosis in patients who undergo splenectomy for malignant hematological diseases and hemolytic anemias.

## Figures and Tables

**Figure 1 medicina-59-01325-f001:**
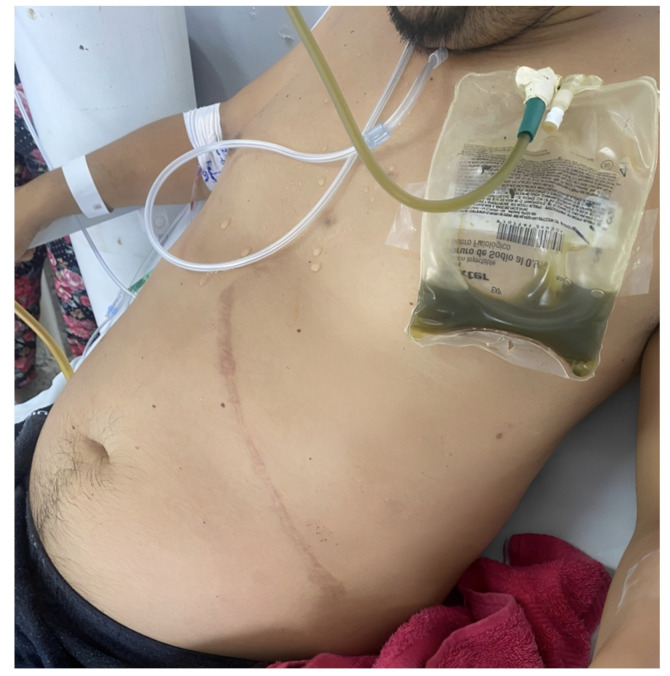
Frontal view of patient with abdominal distension and bilious vomiting.

**Figure 2 medicina-59-01325-f002:**
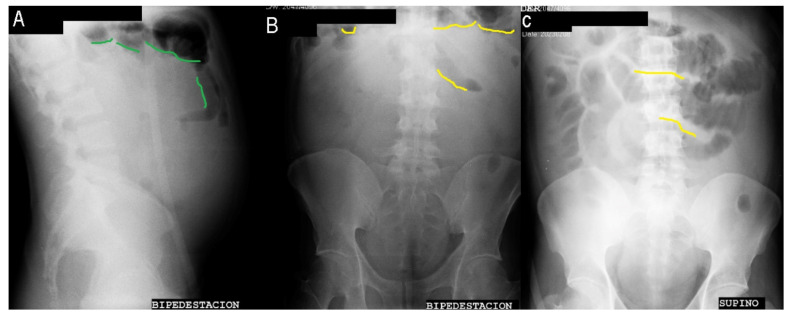
Abdomen series (**A**) lateral view standing abdominal radiography; (**B**) frontal view standing abdominal radiography; (**C**) frontal view X-ray of the abdomen in decubitus supine position with footprint sign. Green color: Pneumatosis intestinalis. Yellow color: Interloop edema.

**Figure 3 medicina-59-01325-f003:**
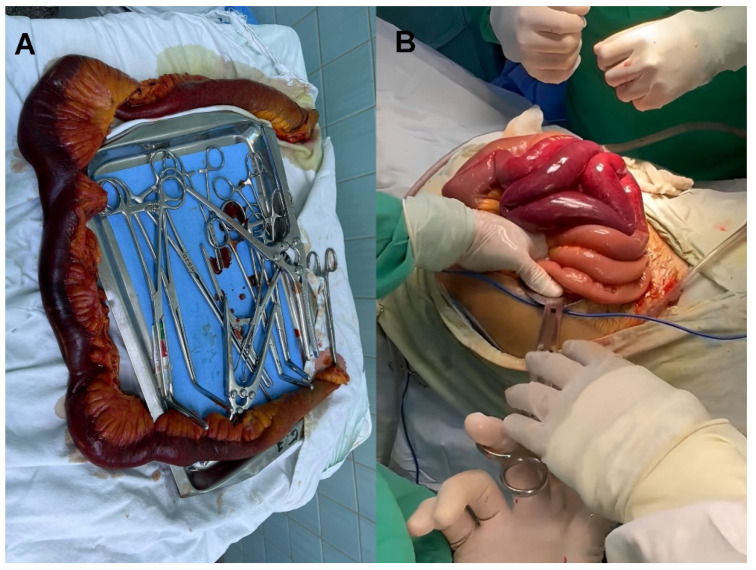
(**A**) Resected part of ischemic jejunum (**B**) Intraoperative mesenteric ischemia jejunum with ischemic tissue.

**Table 1 medicina-59-01325-t001:** Laboratories.

Test Name	Results	Reference Range
Blood count		
- WBC	19.98 × 10^3^/uL	4–10
- RBC	2.25 × 10^6^/uL	3.5–5.6
- HGB	7 g/dL	14–18
- HCT	20.9%	41–50
- MCV	93.2 fL	79–101
- MCHC	33.4 g/dL	31–37
- Platelet count	243 × 10^3^/uL	150–450
- Neutrophils	75.6%	45–65
- Lymphocytes	15.9%	20–45
- Monocytes	6%	0–12
- Eosinophils	2.5%	0–10
- Basophils	0%	0–3
PT	14.1 s	11.5–13.5
aPTT	26.5 s	25–35
INR	1.25	0.9–1.3
Bilirubin, Total	1.29 mg/dL	0–1
Bilirubin, Direct	0.80 mg/dL	0.1–0.3
Creatinine	0.54 mg/dL	0.7–1.3
BUN	0.49 mg/dL	9–23
Sodium	140 mmol/L	135–145
Potassium	3.8 mmol/L	3.5–5.0
Chlorine	104 mmol/L	98–107
HbA1C	4.8%	4.0–5.4
Ketone bodies	0.4 mmol/L	0.6–1.0
Lactic acid	20.2 mg/dL	4.5–19.8
pH	7.415	7.32–7.45
pCO_2_	34 mmHg	35–38
PO_2_	73.4 mmol/L	80–100
CHCO_3_−	22.4 nmol/L	22–26
O_2_SAT	95.1%	94–98
Blood culture	Negative	120 h of aerobic incubation.
Urine culture	Negative	48 h incubation
Coprological	positive occult blood	

## Data Availability

Not applicable.

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
