# Peer review of "Mesenteric Ischemia in a Splenectomized Patient with Auto-Immune Hemolytic Anemia: Case Report"

_medicina, 2023, doi:10.3390/medicina59071325_

Round 1
Reviewer 1 Report
Title
Mesenteric ischemia in a splenoctomized patient with auto-immune hemolytic anemia: Case report
Comments
This manuscript is discussing an interesting topic .Some points need to be addressed.
The abstract is too long.
The author should stress the new information that is added to the literature by this case.
According to the abstract this is a common association. So, what can this manuscript add?
29.. from 5% to 50%, and early diagnosis and treatment can decrease its mortality rate from 5% to 0%. ...English editing and grammar revision are needed.
Appropriate citing of references is mandatory as much knowledge without references.
37… Some studies ...some studies and only one reference at the end
50.. significant splenectomy.…. What is meant by this strange term?
148.. There have been a few reports....it will be better to add references.
149…hemolytic disease…it is a strange term ..the authors meant hemolytic anemia
152.. In the case of our patient. English editing and grammar revision are required.
217.. There is little information in the literature on patients with splenectomy who present 217 mesenteric ischemia…this can be considered as a research gap and will be better to be transported to the end of the introduction section .

Moderate editing of English language required
Author Response
We appreciate your comments and corrections, which have been taken into account to improve the manuscript.
This manuscript is discussing an interesting topic. Some points need to be addressed.
The abstract is too long.
Response: abstract was modified.
Abstract: The author should stress the new information that is added to the literature by this case.
Response: line 14-18 it is emphasized.
29.. from 5% to 50%, and early diagnosis and treatment can decrease its mortality rate from 5% to 0%. ...English editing and grammar revision are needed
Response: corrected on line 26.
Appropriate citing of references is mandatory as much knowledge without references.
Response: It was corrected.
37… Some studies ...some studies and only one reference at the end
Response: It was corrected.
50.. significant splenectomy.…. What is meant by this strange term?
Response: It was corrected in line 51.
148.. There have been a few reports....it will be better to add references.
Response: It was corrected
149…hemolytic disease…it is a strange term ..the authors meant hemolytic anemia
Response: It was corrected on line 144
152.. In the case of our patient. English editing and grammar revision are required.
Response: It was corrected
217.. There is little information in the literature on patients with splenectomy who present 217 mesenteric ischemia…this can be considered as a research gap and will be better to be transported to the end of the introduction section .
Response: It was corrected

Reviewer 2 Report
Per attached file

Substantial editing needed to improve the reading.
Author Response
We appreciate your comments and corrections, which have been taken into account to improve the manuscript.
Science
The authors present the various lab tests in Table 1. What is surprising is that apparently
no coagulation or hemostatic tests are reported. Not even the PT (prothrombin time) or
aPTT (activated partial thromboplastin time). For a patient who is likely to be put on
anticoagulants, one would expect such information to be routinely sought, and to be
presented.
Response: included in the table and in the interpretation in the text.
It seems reasonable to ask why it is that some patients with history of splenectomy
following hemolytic anemia have subsequent thrombotic complications while others do
not. Is it possible that the vulnerable patients have additional predisposing factors? Some
such examples may include: APC resistant Factor V (aka: Factor VLeiden), TFPI deficiency
(± Protein S deficiency), Prothrombin G2021A mutation (ï‚ prothrombin), ï‚ PAI-1, etc.
While it may well be difficult for some clinical labs to identify all known thrombophilic
conditions, that doesn’t make them any less real. It would be helpful to include at least
some mention of these in the discussion of potential added contributing factors.
Response: added on lines 191 -195.
Presentación
- Change “ischemia in a splenoctomized patient” to “ischemia in a splenectomized
patient”
Response: corrected
43-45: Change “case report was De-identified patient specific information, meaning that all data that could individually identify the patient were removed.” to “case report was deidentified. All patient-specific information, that could individually identify the patient, was removed.”
Response: corrected
50-54: Change “A 33-year-old Colombian man with a history of significant splenectomy seven years ago due to autoimmune hemolytic anemia and insulin-requiring diabetes mellitus type 2 three years ago, was referred to our institution on February 4, 2023, for presenting a clinical picture of one month of evolution consisting of colicky abdominal pain in the right hypochondrium associated with consumption of cholecystokinin food.” to “A 33-year-old Colombian man was referred to our institution on February 4, 2023. Over the past month he has been developing colicky abdominal right hypochondrium pain, associated with consumption of cholecystokinin-rich food. His prior history includes: a) as a result of autoimmune hemolytic anemia he had a splenectomy seven years ago, and b) he was diagnosed with insulin-requiring diabetes mellitus type 2 three years ago.”
Response: corrected
54-57: Change “However, two days before, he began to present colicky abdominal pain in the upper hemiabdomen, not irradiated, with a pain assessment scale of 10/10, accompanied by nausea and multiple emetic episodes of bilious content.” to “Two days previously, he began to have intense (10/10) non-radiating colicky abdominal pain in the upper hemiabdomen, accompanied by nausea and multiple emetic episodes of bilious content.”
Response: corrected
65: Change “As for his surgical history, he had a successful splenectomy” to “He had a successful splenectomy”
Response: corrected
66-68: Change “the patient showed signs of viability with BP 115/82 mmHg, HR 102 bpm, FR 16 rpm, T° 36.4 °C, and was diaphoretic.” to “the patient was diaphoretic with these physical signs: BP 115/82 mmHg, HR 102 bpm, FR 16 rpm, T° 36.4 °C
Response: corrected
82: Change “ketone bodies within normal limits and negative blood and urine cultures,” to “ketone bodies were within normal limits. Blood and urine cultures were negative.”
Response: corrected
83-84: Change “Coprological: coffee of liquid consistency, positive occult blood, increased bacterial flora, no intestinal parasites are observed, lipase 0.55 (see Table 1.)” to “Coprology: coffee-colored with liquid consistency, positive for occult blood, increased bacterial flora, negative for intestinal parasites, lipase 0.55 (see Table 1).”
Response: corrected
154-155: Change “Mesenteric vein thrombosis has a predilection for the small intestine rather than the colon.” to “Mesenteric vein thrombosis tends to affect the small intestines more frequently than the colon.”
Response: corrected
221: Change “that a series of abdomen is a diagnostic support,” to “that an abdominal series is helpful for diagnostic support.”
Response: corrected
222: Change “and in these patients, after hours, they will present ascites due to extravasation of liquid.” to “After some hours, these patients will present with ascites due to fluid extravasation.”
Response: corrected
Table 1: i. ‘Blood count’ is an appropriate heading for the various blood cells. However, it is not appropriate for bilirubin, electrolytes, blood gasses, cultures (blood/urine) or coprology. These should have their more appropriate headings. ii. Change “Keton bodies” to “Ketone bodies”
Response: corrected
